# 'People like you?': how people with hypertension make sense of future cardiovascular risk—a qualitative study

Iain J Marshall, Charles D A Wolfe, Christopher McKevitt

Population Health and Environmental Sciences, King's College London, London, UK

**Correspondence to**
Dr Iain J Marshall;
iain.marshall@kcl.ac.uk

## ABSTRACT

**Objectives** Cardiovascular disease (CVD) prevention guidelines recommend that patients' future CVD risk (as a percentage) is estimated and used to inform shared treatment decisions. We sought to understand the perspectives of patients with hypertension on their future risk of CVD.

**Design** Qualitative, semistructured interviews and thematic analysis.

**Participants** People with hypertension who had not experienced a cardiovascular event recruited from primary care.

**Setting** Participants were purposively sampled from two primary care practices in South London. Interviews were transcribed, and a thematic analysis was conducted.

**Results** 24 people participated; participants were diverse in age, sex, ethnicity and socioeconomic status. Younger working-aged people were under-represented. Contrasting with probabilistic risk, many participants understood future CVD as binary and unknowable. Roughly half of participants avoided contemplating future CVD risk; for some, lifestyle change and medication obviated the need to think about CVD risk. Some participants identified with one portion of the probability fraction ('I'd be one of *those* ones.'). Comparison with peers (typically partners, siblings and friends of a similar age, including both 'healthy' and 'unhealthy' people) was most frequently used to describe risk, both among those who engaged with and avoided risk discussion. This contrasts with current risk scores, which describe probabilities in people with similar risk factors; many participants did not identify with such a group, and hence did not find these probabilities meaningful, even where correctly understood.

**Conclusions** Risk as typically calculated and communicated (eg, the risk of '100 people like you') may not be meaningful for patients who do not identify with the denominator. Comparing an individual's risk with their peers could be more meaningful.

## INTRODUCTION

Cardiovascular diseases (CVDs) comprise coronary artery disease, peripheral vascular and aortic disease and stroke and are the top cause of death globally.[1] Deciding about whether to start treatment to prevent CVD involves weighing up finely balanced benefits and harms.[2–4] Current guidelines recommend that clinicians produce an estimate of

### Strengths and limitations of this study

► This study used qualitative methods to better understand the perspectives of people with hypertension on their future risk of developing cardiovascular disease.
► The participants were diverse socioeconomically, and concerning ethnicity, country of origin, age, sex and length of time diagnosed with hypertension.
► We were unable to recruit any participants under the age of 50 years, which may limit the generalisability of the results.

a person's *global risk* (ie, levels of multiple risk factors are measured and combined to produce a risk estimate using an algorithm often embedded in the electronic patient record).[5–7] This risk estimate is most often expressed as the percentage probability of developing CVD in the subsequent 10 years. There is a large body of evidence about how to communicate risks most effectively. Among others: strategies such as using natural frequencies with a common denominator, and icon grids (or 'smiley face charts') have been found to make risk easier to understand and are less liable to mislead.[8]

Fewer qualitative studies have examined the perspectives of healthy individuals on their future risk of CVD.[i] A systematic review by Garside and colleagues in 2008 identified nine qualitative studies that discussed perspectives on risk, but all focused on knowledge of risk factors rather than risk in the probabilistic sense.[9] Davison and colleagues published a series of papers describing their qualitative study in South Wales that focused on the causation of CVD.[10–12] The authors described difficulty in translating population-level data to individuals. Specifically, participants in this study had strong ideas of the type of person

---

[i]Here we differentiate between *risk* (meaning a likelihood, probability, or uncertainty) and *risk factor* (meaning the factors, such as hypertension and smoking, which increase the risk of CVD).

who was a 'candidate' for heart disease, typically a person with multiple and markedly high lifestyle risk factors (a fact the authors attribute to the successful penetration of health promotion campaign messages). Each encounter with an individual who does not meet this extreme risk picture (which as the authors note, accounts for most heart disease incidence) is regarded as an anomaly or random incident. Davison argues that health promotion messages focusing on single lifestyle risk factors might paradoxically reduce motivation to change lifestyle, if heart disease is seen as random and inevitable. More recently, Damman and Timmermans conducted a qualitative study of 40 Dutch lay people, examining their understanding of risk factors.[13] They found that participants saw CVD risk as dichotomous (ie, they would either become sick or not) rather than on a continuum. Polak and Green, in their qualitative study of 34 people who had been offered statins, found that few participants considered risk information relevant in their decision making and instead saw treatment as necessary to treat the risk *factor* (ie, their cholesterol level) rather than the risk (their estimated probability of CVD). Bonner and colleagues conducted a qualitative analysis of data from 25 Australian participants who were encouraged to 'talk aloud' when using two CVD risk communication tools.[14] They concluded that participants needed a reference point (example risk probabilities being anchored to 'high risk' and 'low risk labels) to make sense of the probabilities and additionally found that CVD risk estimates derived from a larger number risk factors were felt to be more credible.

This study sought, through in-depth interviews, to investigate patient perspectives on future CVD risk and understand how these might impact treatment decisions. Specifically, we sought to understand the ideas patients with hypertension have about their risk of CVD, the language and concepts patients use in their discussion of risk and how patients respond to example risk information, presented in a similar format to currently available decision aids.

## METHODS
### Patient and public involvement
The study protocol and interview schedule were informed by helpful feedback from the members of the King's College London Stroke Research Patients and Family Group.

### Sampling and data collection
We interviewed patients from two general practices (primary care practices) in South London. Participants were eligible if they had been diagnosed as having hypertension but with no previous diagnosed CVD. Those with secondary hypertension (eg, pregnancy-related hypertension) or diabetes were also excluded, since their experiences seemed likely to differ. Participants were invited to participate voluntarily, provided with study information to read and asked to sign a written consent form if

they agreed to take part. Participants were purposively sampled, aiming for maximum variation in age, ethnic group, a variety of times since diagnosis and approximately equal numbers of men and women.[15] We initially planned to conduct the research at a single practice, but after some of the initial interviews were completed, we added a second practice located in an area with higher socioeconomic deprivation to increase the diversity of participants. Since purposive strategies allow the researcher deliberate choice over recruitment, it is regarded as legitimate (and often good practice) to adapt recruitment strategies over the study. The first practice had approximately 20 000 registered patients; the second practice had around 12 000 registered patients. Both practices were located in South East London and cared for a highly mobile, multiethnic population; both had a large population with hypertension. Potential participants were identified from the practices' hypertension registers, and a maximum diversity sample (based on age, sex, ethnicity, and duration of diagnosis) were invited via a letter from their own doctor. We later changed to face-to-face invitation aiming to increase the diversity of participants. To ensure our study complied with the ethics committee requirements, initial contact had to be from the patient's own doctor, and we do not have access to data on the characteristics of those who declined to participate/did not respond at this stage. The recruitment sites were chosen to ensure that the interviewer and participants would have no prior knowledge of one another.

We piloted and refined a semistructured interview schedule at a meeting of the King's College London Stroke Research Patients and Family Group (see online appendix). All interviews were conducted by IJM, a general practitioner from a practice in a neighbouring borough, who was a PhD student at the time of the interviews and who had undergone training in conducting qualitative interviews. During the first two interviews, the concept of future risk was difficult to discuss; participants appeared to find it too abstract. To overcome this, we developed a hypothetical decision aid (see figure 1). This decision aid was similar in format to existing CVD decision aids, such as the *Statin Choice* decision aid,[16] and followed guideline recommendations about optimal methods for communicating CVD risk.[17] The decision aid was used to introduce and prompt discussion around risk, rather as an evaluation of this decision aid specifically. We showed this decision aid to participants as the last part of the interview to avoid influencing responses to earlier questions. Interviews were conducted until data saturation was reached. Qualitative data analysis typically overlaps with data collection; we transcribed and conducted an initial analysis of data in parallel with conducting the interviews, together with written notes made by the interviewer. Interview transcripts, notes from ongoing interviews and the emerging analysis were reviewed regularly at meetings of the investigators (IJM, CDAW and CM) where the decision about data saturation was made, defined as when no new themes appeared to be generated from interviews.[18]

# Should I start a new medicine (amlodipine) for high blood pressure?

For any one person, we can't be certain whether or not they will have a complication from high blood pressure (a stroke, a heart attack, or dying).

However, we can accurately estimate what will happen to a group of people on average.

Imagine 100 people who are similar to you (same sex, same blood pressure, cholesterol, and smoking status).

**On average, over 10 years...**

| if all 100 people took amlodipine | if all 100 did not take amlodipine |
| --- | --- |
| 19 people would die | 22 people would die |
| 8 people would have a stroke | 13 people would have a stroke |
| 9 people would have a heart attack | 12 people would have a heart attack |
| 15 people would get swollen ankles | 3 people would get swollen ankles |

**Figure 1** Example risk chart provided to participants as the final part of the interviews.

## Analysis

Interviews were audio recorded and transcribed verbatim. We did not seek further feedback from participants on the transcripts, other than seeking clarification of any unclear points during the course of the interview. The transcripts were analysed using thematic analysis as described by Ziebland and McPherson.[19] This is a form of grounded theory, where analytical categories are developed inductively from the data, rather than defined beforehand. Specifically, the interview transcripts were coded line-by-line using Dedoose, a qualitative research web application.[20] Codes were developed iteratively; new codes were added as needed to describe the interview data. Finally, these reports were used to produce a final analysis, in which connections and groupings between codes can be made, and overarching themes understood. Ziebland and McPherson describe this as the *one sheet of paper* (OSOP) method, in which the issues from report texts are identified in depth and written down on a single (often large) sheet of paper together with a participant identifier.[19] The researcher can place themes in closer proximity to others that appear to be related. By doing this iteratively, a map of the themes develops, and lines can be drawn in to represent the connections. The output of this stage is a map of concepts and their relationships, from which the

supporting evidence (the relevant text snippets) may be retrieved. In practice, IJM coded the interview transcripts. CM separately coded a sample, and the coding scheme was discussed and revised in regular meetings of IJM, CM and CDAW over the course of the data collection. It was at these meetings that the decision about data saturation was taken. The OSOP method was done initially by IJM, then refined following discussion with CM and CDAW.

## RESULTS

### Characteristics of participants

Twenty-four interviews were conducted; a description of participants' characteristics is given in table 1. Participants had diverse occupations, including engineers, cleaners, health professionals, teachers and manual labourers. FIfteen of 24 participants had retired from employment. Participants were aged from 51 years to 90 years, with 46% male. Fifty-four per cent were born in the UK, while the remainder were born in Africa (21%), the Caribbean (13%), France, Ireland and the USA (one participant each (4%)). Self-reported ethnicity was 54% white British, 13% white other, 21% black African and 13% black Caribbean. Participants had received a hypertension diagnosis between 2 months and 33 years prior to the interview. Twenty-two participants were prescribed regular medication with widely varying reported patterns of medication taking (ranging from participants who took medication regularly without fail to those who took it rarely, or when it was perceived as needed). One participant had taken medication previously but decided to stop, and one participant was considering whether to start medication following a recent meeting with her doctor.

### Thematic analysis

We summarise our findings as a lay model of risk understanding, as shown in figure 2. Our key finding was that CVD risk probabilities were often understood from an *individual* perspective (rather than, as intended, population probabilities). This led to understandings of risk presentation that widely differed from their intended meaning. Perspectives on risk were informed by participants' understanding of the causes of hypertension and CVD, and often through seeing CVD as a sudden, unpredictable and fatal event. Many participants avoided contemplating future CVD risk, as they found it unpleasant, pointless, that it was not a priority in the context of advancing age or other illness or preferred to focus on positive aspects of life. For those who did engage with risk discussion, risk information was interpreted in a number of unexpected ways. Many participants did not identify with the reference population. For some, risk estimates would not take sufficient account of factors they felt to be important (including both factors ignored by the models and factors not accorded sufficient weight). Some participants felt that since no two individuals could ever be the same, population probabilities could never make sense. Others identified with either the affected

**Table 1** Characteristics of participants

| Interviewee code | Practice | Age category | Sex | Ethnicity | Time since diagnosis |
|---|---|---|---|---|---|
| A | 2 | 70–80 | M | White British | 5 years |
| B | 2 | 60–70 | F | White US | 3 months |
| C | 1 | 50–60 | M | White British | 19 years |
| D | 1 | 60–70 | F | White British | 12 years |
| E | 2 | 60–70 | M | Black Caribbean | 15 years |
| F | 1 | 70–80 | M | White British | 5 years |
| G | 1 | 60–70 | M | White British | 3 years |
| H | 1 | 80+ | M | White British | 33 years |
| I | 2 | 60–70 | M | White British | 3 years |
| J | 1 | 80+ | M | White British | 6 years |
| K | 1 | 60–70 | F | White French | 15 years |
| L | 1 | 70–80 | M | Black African | 30 years |
| M | 2 | 50–60 | F | Black African | 6 years |
| N | 1 | 70–80 | F | White British | 5 years |
| O | 1 | 50–60 | F | Black Caribbean | 2 months |
| P | 1 | 70–80 | F | White British | 10 years |
| Q | 1 | 70–80 | F | White British | 30 years |
| R | 2 | 60–70 | F | Black Caribbean | 15 years |
| S | 2 | 70–80 | F | White British | 7 years |
| T | 2 | 70–80 | M | Black African | 33 years |
| U | 2 | 70–80 | F | White African | 20 years |
| V | 2 | 60–70 | F | Black African | 2 years |
| W | 2 | 80+ | M | White British | 20 years |
| X | 2 | 70–80 | F | White Irish | 20 years |

F, female; M, male.

or unaffected portion of the fraction. Conversely, peer comparison was widely used by participants and appeared to be an acceptable and meaningful way of discussing risk. These themes are presented in detail below.

**Causes and consequences of hypertension**
Most of the participants explained their hypertension by way of conventional biomedical risk factors; the majority mentioned one or more of advancing age, obesity, a family history, poor diet and lack of exercise as contributing. These

factors were widely described as causative of high blood pressure but as causative of complications of high blood pressure.

Several participants who were born outside of the UK attributed their hypertension to their new environment. Many participants identified that stress contributed to their hypertension. For some of these participants, stress was regarded as highly important, and for several, it was regarded as the primary cause of hypertension. For several participants, a stressful life

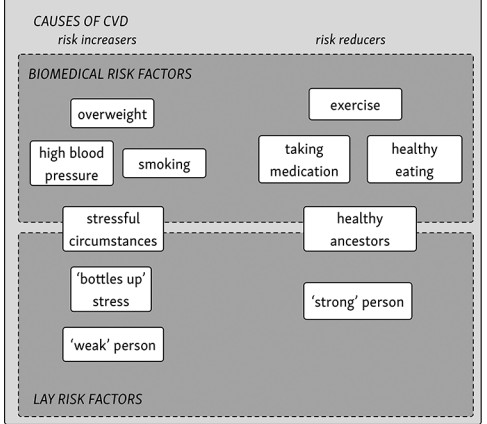
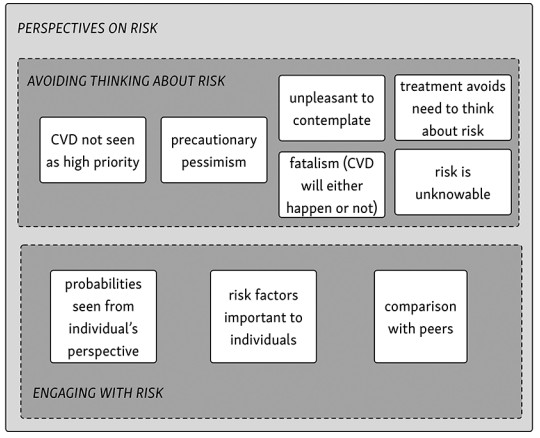
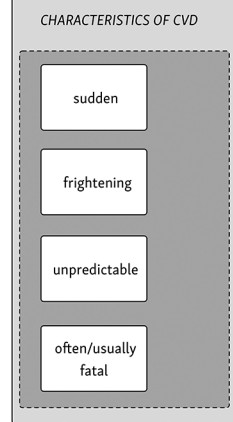

**Figure 2** A lay model of cardiovascular risk. CVD, cardiovascular disease.

event was considered to be a trigger for hypertension starting.

A wide variety of consequences of having high blood pressure were described by participants; these could be broadly grouped as cardiovascular consequences and somatic consequences. Most of the participants described the possibility of developing an illness due to high blood pressure, particularly if not controlled, including stroke, heart attack and kidney disease. Several participants described complications as being sudden, unpredictable and severe; many reported sudden death as an important consequence of untreated hypertension. Several others described sudden increases in blood pressure as harmful, as illustrated by the following two quotes:

> Well I mean I think it can be sort of damaging for the heart as, you know, if it really goes high. And I mean you could get a stroke, you know, a number of things really. (Interview Q)

> At one point, it was extremely high, I borrowed my neighbours cuff, whatever you call it…sphyg… and er, the systolic was over 200 and I got a little frightened… (Interview D)

Several participants described physical sensations they attributed to raises in blood pressure: these included headache and breathlessness. One participant, who attributed physical symptoms to an acute increase in blood pressure, described how the absence of physical symptoms was an important marker of good control.

> I am very careful about it, I have no problem with the blood pressure. If I feel that my head will explode. I relate it with the blood pressure, the only thing I do is I sit down and relax. I breathe deeply, and then with the relaxation it's finished, it's happened one in three months no more. (Interview K)

### Attitudes towards future risk

A large portion of the interview comprised discussion about perceptions of risk. Participants were asked both if they had thought about this previously and also asked whether they considered themselves likely or unlikely to develop a complication of hypertension in future. However, around half of participants did not provide any sort of estimate of whether they would be at risk of hypertension complications themselves, even after prompts.

Many participants expressed a desire to avoid contemplating future risk of disease, including both younger and older participants. Key reasons expressed by participants included: finding the consideration of future serious illness unpleasant or stressful; perceiving that looking at future risk did not make sense due to old age; and having other active health problems that were perceived as more serious making CVD risk less relevant (eg, pain from osteoarthritis).

The concept that contemplating future risk of heart disease or stroke was unpleasant is illustrated by the following quote from a 50-year-old man who worked as

a senior teacher and otherwise saw himself as in good health:

> I don't know whether I would want to know if I'm at risk of stroke, because it would be quite upsetting if I'm at high risk. (Interview C)

Similarly, one participant (who was aged 90 years and continued to live a physically and socially active life and who went swimming daily) attributed his reluctance to consider future risk to a happy life.

> I tell you, it's an amazing life. I don't worry about ill health. (Interview H)

During this part of the interviews, several participants expressed that a key benefit of maintaining good health was that they could avoid thinking about future risk. Another participant (a retired scientist, in his 80s) described the benefits of his healthy lifestyle:

> I have always relied on good health and good living to obviate any need to question these things. (Interview J)

Another participant hinted that he felt saw himself as being at low risk, but also that being on medication meant he did not need to contemplate it further.

> Will I get a stroke? No I don't think I will. I haven't considered it. Because I'm on the medication to stop it. (Interview C)

One participant (who retired early due to poor health) described having chronic pelvic pain and arthritis. For her, CVD was seen as a lower priority than her other health problems:

> I mean I've had a couple of angiograms and my arteries seem to be fine, so I haven't really gone into it. As I say, to me, it's the least of my medical problems, I haven't really… I don't feel like frightening myself with stuff. (Interview D)

### Perspectives on risk probabilities

The example risk chart is shown in figure 1. All participants except one appeared to understand the risk chart's intended meaning and from their discussions and questions appeared to understand the statistics presented. The exception was participant V who did not engage in discussion around the decision aid, saying only, 'It's good to take tablets'. Her response gave the impression she had found it difficult to understand both the text and statistical contents of the tool.

Participants were mixed in their opinion about risk estimates, with around half expressing a strong desire to have such information and around half stated that they would not find it useful.

Several participants expressed surprise at the content of the risk tool, explaining that they expected medication to have a bigger effect. This is illustrated by one participant

who expected that medication would abolish the risk of stroke altogether:

> I was given the impression that it was 'take that and you'll be fine' 'take that and you won't have a stroke…' but not according to this [pointing at example decision aid]… (Interview C)

One participant, surprised by what she saw as a small benefit of treatment, wondered if it would lead people to stop their treatment:

> I mean I realise it's double which is a lot but I thought it would be more like three or four times…So I tell you what it would do, is make me think oh it doesn't seem to make all that much difference whether I take the tablet or not… If I had side-effects from it and I saw that, I'd think well maybe it isn't worth it or maybe it's not worth it all the time, maybe I can skip it when I'm on vacation. (Interview B)

All participants were asked hypothetically whether the absolute benefits the example risk presentation (if they applied to them) would change their decision to take medication (since almost all took medication, this usually meant deciding to stop). None of the participants, including the two who were worried about others stopping, answered that they would stop treatment themselves.

One participant explained that, for her, even a small absolute benefit of treatment would be worth taking:

> Well yeah…cos if four people die here, and six people die here…you'd want to take the tablets. It's about living longer isn't it. (Interview S)

### Risk factors from the individual perspective

Although the risk chart stated that the 'at risk' reference population was similar to the hypothetical patient in terms of known risk factors (age, sex, ethnic group, smoking history and cholesterol levels), many participants expressed a view that statistics such as the example could not represent their own risk. For these participants, risk factors outside of what is conventionally measured were highly important, and their absence led them to see the charts as unreliable and that they would not apply to them personally.

Several participants felt that even where a particular risk factor had been incorporated into the risk score, it had not been accorded enough importance. This is illustrated by a participant who moved to the UK from Nigeria, who perceived that his particularly healthy family history would be strongly protective.

> It's not in my lineage. We never suffered… just go. My senior brother just died 2 years ago. And he never had any illness until just once. My father was the same. My father lived to one-twenty years old, and I understand he never suffered any illness. But that time he just go. What I have in mind about will happen to me. I've made up my mind that that is what will happen to me. Of course. (Interview T)

One participant felt that the risk tool would not have taken into account his (relatively) healthy lifestyle:

> Yeah, but if it could be more personalized. Because I don't smoke, I hardly drink, I mean I do drink, I'm not a monk, but it kind of goes in phases. (Interview C)

Several participants went further, and since individuals are unique, questioned whether a group could ever be regarded as similar, as illustrated by one participant:

> it does say 'imagine a hundred people who are similar to you' but you see, who is? no person is similar to…you know… (Interview I)

### Seeking individual meaning from population statistics—'I think I might be that one'

Several of the participants reported a pre-existing idea of their likelihood of developing CVD and reported that data from risk estimates was unlikely to change that view. This certainty is illustrated by one participant, a woman from Nigeria who saw herself in good health, who expressed disbelief after seeing the example chart, feeling already convinced of outcome of not using blood pressure treatment:

> If like me, I don't take treatment, someone will die, not like straight away, they die, it's not having stroke, the person will just die. (Interview M)

However, several participants from a variety of backgrounds seemed to identify with the affected part of the fraction. For example, the decision aid shown to the participant described that, if not taking amlodipine, 22 out of 100 people would expect to die. These participants thought that they would likely be 1 of the 22 (rather than the intended meaning, of having an equal probability of being any one of the 100). This is illustrated by one participant, using nearly identical wording to several others:

> One person in a hundred is still one too many, and I'd be one of those. (Interview U)

For a few, this idea appeared to be precautionary. It seemed preferable to them to assume they would be affected by a complication in order that they could take any necessary action to prevent it. This is subtly different from perceiving they would certainly be affected. This is illustrated by a UK-born participant who wanted to prepare for the worst case:

> Well, I think that I might be that one [laughs] or one of those, if I stopped taking it, that [I] would die from it or would have a stroke, so, you know, I would rather take the tablets than take that chance. … (Interview F)

However, this idea did not seem to be precautionary for all, and several participants appeared to perceive a certainty about developing or not developing illness. Several of the participants expressed the idea that certain

people are 'stronger', that is, more resistant to illness than others. A weak person would be susceptible to illness in general (not only CVD). A participant who moved to the UK from Nigeria described how 'strong' people would be more likely to survive stroke:

> If you don't die, because sometimes some people are strong in their body, you have stroke. (Interview M)

Another participant (a UK-born, retired man who was not taking treatment) used the same 'strong' and 'weak' terminology, perceiving that weaker people were likely to die from hypertension. He contrasted himself with his wife whom he perceived as weaker:

> I think there are certain people, I keep referring to me, there's certain people who are stronger and some are weaker. You know, my wife's very weak, she would take any tablet you gave her for anything; she wouldn't take two a day but she'd take one a day of those tablets, you know what I mean, and if they said she had chronic whatever she would read about it and say 'Yes, yes, I've got this, this sounds [like] me. (Interview A)

Similarly, a UK-born participant (a retired health professional) described that her sister, by living an unhealthy lifestyle, had become susceptible to illnesses in general:

> She didn't really look after her body much. She didn't do any exercise, it was all sitting down. And she's had a stroke, she had a stroke before the age of 80. She's now… she's 82…she's very sedentary, she's surprisingly upbeat for having had a stroke…she's too overweight, and so on. I just feel she's constantly having little infections. (Interview P)

### Comparing risk with peers

Several participants talked about their risk of by comparing themselves to peers or family (illustrated by the quotation from interview P, above). One participant (who had training in statistics) reported his own risk in a similar way.

> I'd say probably in the top five to ten percent of people of my age, myself, well I've always played sport, played rugby until I sort of got too old and too crocked and knocked and played squash ever since… (Interview I)

One participant explained the fact she had not developed complications from hypertension by comparing her own response to the diagnosis with that of her father-in-law's; her own reaction included obtaining and taking medication, addressing being overweight through bariatric surgery and keeping as active as possible.

> My father in law was diagnosed as having quite high blood pressure, and he turned himself into an invalid overnight practically, he did nothing, and it made matters worse as he died suddenly of a heart

attack… So that's a completely different reaction to mine I think. (Interview C)

### DISCUSSION

This study sought to examine experiences and ideas about cardiovascular risk from the perspective of people with hypertension. A key theme was the concept of patients as individuals. Notably, no one who was interviewed described their own risk in terms of probability. Around half of the participants expressed doubt that a risk communication tool could apply to them personally. Many of these participants felt that the use of the phrase 'people who are similar to you' was of dubious reliability, since they felt the reference group would not be sufficiently similar to them. For some participants, ensuring that statistical estimates took adequate account of factors they held important would lead them to be useful. However, for many, no population statistic would be applicable to them as individuals, since no two people are exactly alike. Similarly, several participants did not interpret the risk presentation as probabilistic, but instead pointed at the numerator of the fraction saying, 'I'd be one of those ones'. Participants widely expressed their risk in terms of their peers: typically siblings or friends of a similar age.

Many of the participants expected that medication would be highly protective; several expressed surprise at the small size of benefit displayed on the risk chart example; one of whom expressed disbelief. Several participants expected medication would provide absolute protection against complications from hypertension. This mirrors the results of quantitative analyses of patient expectations of treatment for a number of conditions; a systematic review found the majority of 27 323 participants of 35 studies overestimated the benefit and underestimated harm from a wide range of medical treatments.[21]

Parallels may be drawn between many of the themes raised in this study and debate about the problems with risk communication raised in the academic literature.[8] The concerns expressed by participants echo the *reference class problem*.[22] Given that individuals have an unlimited number of characteristics and can potentially belong to many populations, which population does the risk estimate describe? Several existing decision aids do report a clear denominator (and making a denominator clear is a key recommendation in guidelines on the production of decision aids),[17] but some do not make this distinction. The heart-to-heart decision aid describes cardiovascular risk using the following form of words:

> your risk of having a cardiovascular event in the next 10 years is 11%.[23]

One difference in our study is that the example risk description did have a clearly stated reference population (stated as 'Imagine 100 people who are similar to you (same sex, same blood pressure, cholesterol, and smoking

status'). Here, all but one of the participants seemed to understand the reference category, but most perceived it to be irrelevant. Many participants expressed that their personal characteristics could not be adequately incorporated into a statistical estimate.

For many of the participants, risk estimates could not sufficiently take account of the particular risk factors that they perceived to be most important. Emmons and colleagues reported similar findings in their focus group study, which examined patient perspectives on the Harvard Cancer Risk Index (a tool that presents a personalised estimate of cancer risk to patients based on their weight, smoking, alcohol consumption, family history and history of inflammatory bowel disease).[24] They found many participants held certain factors as important that were not incorporated by the tool (including poverty, air pollution and exposure to toxic waste). The absence of these factors caused these participants to view the final risk score with some scepticism. The qualitative study by Damman *et al* found that participants reported similar risk factors for CVD as experts but that they assigned them different values: an unhealthy lifestyle (and particularly high levels of stress) was seen as the predominant cause by the lay participants, whereas experts gave similar weighting to advancing age and genetics.[13] Bonner *et al* in their 'talk-aloud' study found that CVD tools that accounted for a greater number of risk factors were seen as more credible.[14] Our study suggests that the *number* of risk factors might be less important than whether the risk factors used match those that patients perceive as important.

However, the choice and weighting of variables in CVD risk algorithms is guided more by statistical and pragmatic concerns. For example, QRISK selected risk factor variables that are routinely collected in electronic health records. This allowed the use of a vast dataset (health records from 11 million patients) to derive model parameters[25] and also allows clinicians to calculate risk estimates in the consultation using data that often already exists on the patient record and without the need for onerous collection of additional data.

The trade-off of this approach is that the model is highly dependent on the variables that happen to have been collected. In UK primary care records, blood pressure, cholesterol levels and smoking status are readily available with good data completeness. Diet and physical activity, although both recognised as important risk factors, are not routinely collected in a standardised way (and are intrinsically more complex to quantify). The participants' concerns that risk estimates did not take into account their lifestyle changes are therefore valid.

Of the participants of this study who did express an idea of their own risk, all used comparisons with peers and family members. This used spontaneously by most of the participants and including a number of those who wished to avoid consideration of risk in a probabilistic sense. Interestingly, this method also overcomes the reference class problem. By comparing themselves to peers (typically friends and siblings close in age), participants were able to have an understanding of their risk as an individual.

Although most current CVD risk tools generate an individualised percentage risk, the Dundee Heart disease rank (one of the earliest methods for calculating a global CVD risk) does evaluate an individual's risk in comparison with their peers. Using this approach, individuals are assigned a position in a 'coronary queue' of 100 people.[26] The 'queue' is designed to represent the general population of the same age (or age group) as the target individual. The queue position was determined by reference to the risk factors of 10 359 men and women aged 40–59 years in the Scottish Heart Health Study. This reference group, like the peers and family members described here, were not differentiated by risk. Current standard risk estimate percentages (eg, as those generated by QRISK2 or the Framingham score) could be translated to a population rank, given knowledge of the risk estimates of a sample of the general population in a particular age group. Risk calculation tools are widely used, and therefore population data should be easy to collect, for example, from analysis of routine primary care records data. We provide an illustrative example of how such an approach might work in figure 3. One disadvantage is that this approach still ultimately relies on conventional risk models (which still often do not include the variables participants hold to be most important). Nonetheless, such a presentation could overcome some of the comprehension issues identified in this study and is worthy of further investigation. Most empirical studies of CVD risk formats have focused on presenting population absolute risks,[27] and though a small number of studies have examined the provision of a population reference risk as a comprehension aid,[28] none (to our knowledge) have assessed how patients respond to a rank system.

### Strengths and limitations of the study

We were largely successful at recruiting people of a wide range of geographic, ethnic and socioeconomic backgrounds and with a wide range of durations since diagnosis and recruiting both men and women. People of a wide range of ages were recruited; participants ranged from 51 years to 90 years, corresponding roughly with the age-specific disease burden found in population studies. However, we were unable to recruit anyone younger than aged 50 years, despite offering the opportunity for evening or weekend interviews (the expected prevalence of hypertension for those aged 40–49 years is around 20%, although most this group will be either unaware or aware but untreated).[29] This group may have different attitudes towards risk and information than the older age groups. Key themes (including not wishing to contemplate future risk) did occur in interviews were not limited to older participants and also arose in interviews with participants in the younger part of the range. Similarly, there was no obvious impact of ethnicity or country of origin (although overall

Consider a group of 20 men, picked at random, of about your age.
We estimate that you would be the 3rd most likely to develop a heart attack or stroke.

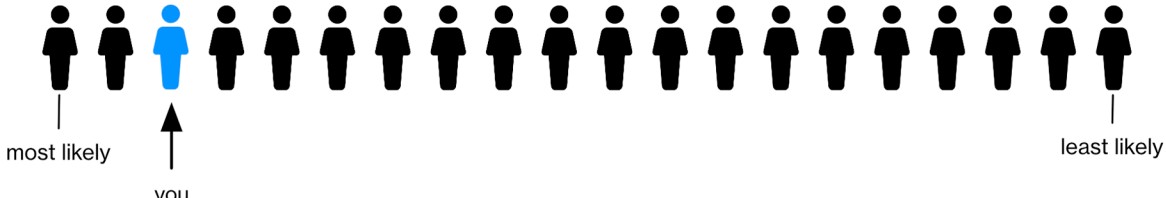

If you chose to take medication for blood pressure, your risk of heart attack and stroke would reduce.
We estimate you would move to 8th place.

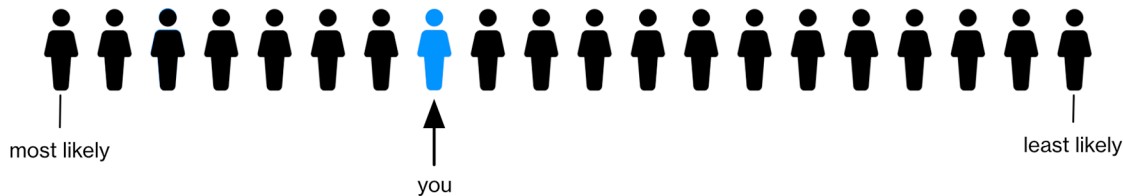

**Figure 3**  Illustrative example of a queue or ranking system for communicating cardiovascular risk.

numbers were small to determine such a link): the key themes were repeated in participants of a variety of backgrounds.

The dual role of a clinician as a researcher may affect participant responses and make it difficult to remain impartial.[30] In this study, there is a risk that participants who were aware that the interviewer was a general practitioner would be less likely to talk openly about medication non-adherence and be reticent to reveal views they perceive the interviewer would disapprove of. Similarly, as a requirement of the ethics process, the initial approach had to be from the participant's own clinical team. We aimed to avoid the appearances of a clinical consultation as much as possible (the interviewer introduced himself as a PhD student, dressed casually and the interview room was not set up as a consulting room). We additionally made clear both on written information, and verbally at the beginning of interviews, that all information remained confidential and would not be shared with their doctor. It is possible, nonetheless, that participants were reluctant to talk openly about decisions to avoid prescription medication.

It is possible that our example decision aid (which was designed similarly to many others used in practice) influenced participants responses. To limit this as much as possible, we showed participants the aid as the final part of the interview. Additionally, many of the participants responded to the information in the aid in unexpected ways (eg, identifying themselves with the affected portion of the risk fraction, rather than interpreting as a probability).

## CONCLUSIONS

Risk communication requires the successful transmission of a statistical probability and the the communication of the context and meaning of the number. For those who want to engage in discussions about risk, tools that incorporate risk factors that are important to patients may be perceived as more meaningful. However, for many, risk in the format commonly used in current decision aids and predictive tools (ie, explicitly or implicitly presenting the risk of '100 people like you') was not seen as relevant, since they did not identify with the denominator. Some did not believe that any reference group could be sufficiently like them and sought risk information that would apply to them as an individual. Ranking an individual's risk against their peers may be more acceptable and relevant than conventional probabilistic approaches.

**Acknowledgements**  We would like to thank the King's College London Stroke Research Patients and Family Group for their valuable comments that helped improve the clarity of language used in the interview questions. We would like to acknowledge the enormous help of the two GPs who allowed us to conduct the study in their practices and facilitated recruitment. Finally, we would like to express gratitude to the patients who freely gave their time for interviews.

**Contributors**  Design of study: all authors. Selection and recruitment of participants: IJM. Conducting interviews: IJM. Initial coding of interview transcripts. IJM and CM. Refining of coding framework and thematic analysis: all authors. Initial draft of manuscript: IJM. Review and editing of multiple versions and sign off of final version of manuscript: all authors.

**Funding**  IJM is supported by UK Medical Research Council (MRC), through its Skills Development Fellowship programme, grant MR/N015185/1. The research was funded by the National Institute for Health Research (NIHR) Collaboration for Leadership in Applied Health Research and Care South London at King's College Hospital NHS Foundation Trust (award number NIHR CLAHRC-2013-10022), Prof Charles DA Wolfe (stroke theme lead), http://www.nihr.ac.uk/about-us/

how-we-are-managed/our-structure/infrastructure/collaborations-for-leadership-in-applied-health-research-and-care.htm.

**Disclaimer** The views expressed are those of the authors and not necessarily those of the NHS, the NIHR or the Department of Health and Social Care. The funders had no role in study design, data collection and analysis, decision to publish or preparation of the manuscript.

**Competing interests** None declared.

**Patient consent** Not required.

**Ethics approval** The study was approved by the Proportionate Review Sub-committee of the NRES Committee London, Wandsworth Research Ethics Committee.

**Provenance and peer review** Not commissioned; externally peer reviewed.

**Data sharing statement** There are no additional data available from this study.

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
