## [Reviewer comments · BMJ Open]

ARTICLE DETAILS

TITLE (PROVISIONAL)	“People like you?”:how people with hypertension make sense of future cardiovascular risk—a qualitative study
AUTHORS	Marshall, Iain; Wolfe, Charles; McKevitt, Christopher

VERSION 1 – REVIEW

REVIEWER	José Pablo Werba Centro Cardiologico Monzino, IRCCS, Milan, Italy
REVIEW RETURNED	15-May-2018

GENERAL COMMENTS	In the title, the reason why the authors used the expression: People like you? is not evident. In addition, some information about the type of population included in the study should be appropriate (hypertensive patients). Is the research question or study objective clearly defined? :No The authors describe what seem(s) to be the objective(s) of the study in the last part of the introduction through three groups of questions (first, page 4, lines 27-31; second, page 4, lines 34-39; third, page 4, lines 39-44), which are connected using logical links. Therefore, the reader may not directly apprehend which are the primary or secondary objectives of the study. Is the abstract accurate, balanced and complete?: No In the results section, the themes identified in the analysis of the data are not indicated, as reported in Methods/Analysis (page 6, lines 21-57) and the results are presented as a narrative, without a clear framework as commonly performed in qualitative studies. Are the methods described sufficiently to allow the study to be repeated?: No 1-The semi-structured interview schedule piloted and refined by the researchers is not provided (though indicated its existence in the online Appendix). 2- Some criteria of COREQ guidelines for reporting qualitative studies are lacking such as: previous knowledge between the interviewer and the participant, duration of the interviews, exposure of the transcript to the interviewer for comments or corrections and features of candidates who refused to participate in the study. Are research ethics (e.g. participant consent, ethics approval) addressed appropriately?: No The approval by ethic committees is described (page 5, lines 14-19) but whether / how the participant provided informed consent is not specified. Do the results address the research question or objective? No
---

	The research question/s was/were ill-defined and therefore, it is hard to say whether the results address them properly. In particular, the authors indicate that “the transcripts of the interviews were analysed using a form of grounded theory where analytical categories are developed inductively from the data rather than defined beforehand”. Codes and themes extracted from the transcripts should answer predetermined research questions related to the main topic of the study which is “cardiovascular risk”. Instead, the thematic analysis encompasses some aspects not related to the “patient perspectives on cardiovascular risk”, such as, for example, causation of hypertension. Are they presented clearly? No 1- In page 7, lines 44-51, the authors describe the level of literacy of participants about causes of hypertension and not their perception of the risk that overt hypertension entails. This information might have been interesting if the impact of literacy on the perception of risk had been evaluated. 2- Some of the meanings derived from the transcripts do not seem to reflect adequately the participants’ words. For example, in page 9, lines 29-37, the authors’ interpretation is that “a key benefit of maintaining good health was that (they) could avoid thinking about future risk”, whereas the actual expression of the participant suggests that he/she has other medical problems that make futile for him/her the risk of complications of hypertension. Another examples are found in page 10, lines 19-21 and in lines 37-39 which actually do actually not depict reluctance to consider future risk. Indeed, the first participant, right or wrong, considers that he/she has a low chance of stroke (therefore, he already considered that risk) while the second one is expressing at least two potential consequences of hypertension (heart damage and stroke). Conversely, reluctance to risk estimates is described out-of-place in the paragraph of “Response to example risk charts” (page 11, lines 7-11). 3- It is appropriate to transcribe verbatim the participants’ words. However, it may be somewhat forced to obtain meaning units from incomprehensible sentences such as, for example, that in page 13, lines 21-23but it kind of goes in phases (?). In other words, the authors would select for the analysis only phrases that convey a well identifiable meaning. Are the discussion and conclusions justified by the results: No In conclusions, the idea proposed by the authors to engage patients in discussions about risk of using tools that incorporate risk factors for complications of hypertension which are regarded important to patients may be interesting but not real. In fact, available risk algorithms do not include some features, hardly measurable, that patients may consider relevant, such as family history, exposure to stress, dietary habits, etc. In addition, though It is not obvious how could an estimate of risk could be objectively and scientifically provided by comparing a subject’s risk with that of his peers or family members, which should be itself estimated somehow. Is the supplementary reporting complete (e.g. trial registration; funding details; CONSORT, STROBE or PRISMA checklist)?: No The authors did not mention the utilization of any checklist for reporting qualitative research such as COREQ (COnsolidated criteria for REporting Qualitative research). In addition, as indicated previously, the interview schedule was not provided. Is the standard of written English acceptable for publication? No There are many phrases throughout the text without an acceptable written English. Some of them are actually unintelligible. Therefore,
--	---

	we suggest the authors to subject the manuscript to a mother tongue scientific writer check before re-submission to any journal.
--	--

REVIEWER	Yeunjung Kim Yale University School of Medicine. CT, USA.
REVIEW RETURNED	18-May-2018

GENERAL COMMENTS	This manuscript focuses on the much-needed topic of decision aids and understanding of personal CV risk and future risk. There were many aspects of the study which were enlightening and fortifying known data. I do appreciate the qualitative aspect of the study and I believe authors have intimately detailed the discussions between the interviewer and interviewees. They are using a verified and stringent method for building a narrative of the qualitative data. The OSOP method is described in general but it is unclear exactly how data saturation was met. This process could be a bit more transparent because the point of data saturation can definitely impact the outcome of a study this size. In summary, at this time there is a lack of integration of qualitative findings to provide a rich narrative mostly due to the structure of the presentation. I think that the manuscript could be better organized to present the rich details of the interviews. There are few errors in the manuscript/abstract which can easily be fixed.  - Patient characteristics: How about including comorbidities which may be associated with poor outcomes from hypertension (such as diabetes or coronary artery disease)? - Table 1 could be more detailed (i.e. add education level and/or socioeconomic status). Table 1 could show the proportion/mean of each patient characteristics (i.e. mean age, race, socioeconomic status, education level, etc.) - Rather than listing simple ideas, if ideas can be grouped into certain overarching themes or concepts, this would be much more effective. (This can be applied throughout the results section.) For example, the “lay model” of cardiovascular risk could be presented initially and supported in the results section with the participant quotes. - Please integrate and present overarching findings. For example, there was one patient who felt reluctant to consider future risk because of “a happy life.” Rather than just having a quote saying that, by providing a more complete detail about the patient’s personal background, the readers may understand why he would say that (aside from being of older age). I think it would make the overall paper stronger. This is a general to the paper as a whole. - Regarding the cause of hypertension. Unfortunately, much of the qualitative responses appear to be short answers rather than detailed narratives. At least, this is how the authors have represented the results. In this case, it may be more valuable to quantify the responses. I wished for more details of the respondents for which contextualization was done. For example, in attributing hypertension to new environment and stress, were there similarities of the participants aside from being born outside of the UK? Were the participants diagnosed with hypertension in the UK or prior to coming to the UK? Is there any association between how patients answered this question and how they applied the future CVD risk from hypertension? The vague descriptors (such as more than half, several, most, large portion, around half, many) were distracting
--

	especially when it was done in succession (i.e. in the paragraph regarding stress as a contributor to hypertension: First sentence says more than half of the participants. The next sentence says for some of these participants. The next sentence starts for several participants.) - Regarding the reluctance to consider future risk. This was especially an important finding. As we know, the ability to perceive risk may be important in modifying behavior. Please define how “many participants expressed desire to avoid contemplating future risk.” Who were these people versus participants who were more engaged in considering future risk? Here, you may say that the avoidance was due to 3 themes: 1) inability to see the urgency of the risk, 2) purposeful neglect by focusing on the positive, and 3) belief that complications are unavoidable. This may be a stylistic difference in approach but helps us, the reader, to categorize the responses early on. So I believe having the key components of the “lay model” described earlier and supported would be stronger format than using it as a basic summary figure in the end. - Regarding response to example risk charts. The authors can be more direct when describing more quantitative aspects of the study. For example, it would be more clear if authors say this # of participants found risk charts helpful and not just say around half. And who are these people and how are they similar or different to those who disagreed? Discussion and conclusion. - I thought the discussion was very interesting and sets up value in this study. I do believe that patients are actually correct when they find the denominator irrelevant to them. Although using probabilities come naturally to those in academics or in medicine, it makes sense how it may not be intuitive for some. Especially, this may actually impact how patients will perceive his or her risk. Using familiar examples (i.e. friends, family, sibling) makes it relevant to the individual. The inclusion of the ranking model, therefore, is appropriate in the conclusion.
--	---

REVIEWER	Carissa Bonner University of Sydney, Australia
REVIEW RETURNED	28-May-2018

GENERAL COMMENTS	Thank you for the opportunity to review this qualitative paper on CVD risk perceptions amongst hypertensive patients. The paper is generally well written (but with some minor grammatical errors throughout – please read carefully before next submission), describes the analysis methods very clearly, and provides some interesting findings and conclusions. However the paper needs to include much more explanation of the existing qualitative literature on this topic since the cited 2008 review, and how this study addresses a gap or adds to the literature. Is your specific patient population/context different, or does your interview schedule include a new angle? How do your findings relate to other qual studies and reviews on CVD risk perception, CVD decision aids, and recommended risk communication formats for decision aids? Specific suggestions/comments are below. STRENGTHS AND LIMITATIONS BOX Re: “Although many qualitative studies have examined patient perspectives on hypertension, this is one of few studies to examine perspectives on future cardiovascular risk” - There is a lot of
--

literature on future CVD risk perceptions not cited so I don't think this statement really reflects a key strength of the study – many studies looking at this issue include a more comprehensive range of risk factors for CVD (e.g. high cholesterol and/or lifestyle risk factors as well as hypertension) so may have been missed because of that – but they are still relevant.

INTRODUCTION

How is hypertension defined in this context (thresholds vary by country/guidelines), and why have you chosen to focus on this single risk factor for CVD and not a more comprehensive range of risk factors for which blood pressure medication may be prescribed?

There is a lot of qualitative research on the topic of future CVD risk perception since the 2008 review cited, which should be discussed, not just the research cited in South Wales. I suggest looking at the reviews below and many papers in the journal Patient Education and Counseling for a large literature on this topic. Some examples you might look into include:

EXAMPLES OF OTHER QUALITATIVE STUDIES ON FUTURE CVD RISK PERCEPTION

- Qual study in Netherlands – Damman, O. C., & Timmermans, D. R. M. (2012). Educating health consumers about cardio-metabolic health risk: What can we learn from lay mental models of risk? *Patient Education and Counseling*, 89,300–308.
- Qual study in UK - L. Polak, J. Green, Using quantitative risk information in decisions about statins: a qualitative study in a community setting, *Brit. J. Gen. Pract.* 65 (2015) e264–269, doi:<http://dx.doi.org/10.3399/bjgp15X684433>.
- Qual study in Australia – Bonner C, McKinn S, Lau A, Jansen J, Doust J, Trevena L, McCaffery K (2018). Heuristics and biases in cardiovascular disease prevention: How can we improve communication about risk, benefits and harms? *Patient Education and Counseling*, 101(5), 843–853. doi:<https://doi.org/10.1016/j.pec.2017.12.003>

RELEVANT REVIEWS ON CVD RISK PERCEPTION (INCLUDING QUALITATIVE STUDIES)

- Review of CVD risk communication formats including some qual studies - T. Waldron, S. van der Weijden, J. Ludt, G. Gallacher, What are effective strategies to communicate cardiovascular risk information to patients? A systematic review, *Patient Educ. Couns.* 82 (2011) 169–181, doi:<http://dx.doi.org/10.1016/j.pec.2010.04.014>.
- Qual review on reducing CVD risk - Murray, J., Honey, S., Hill, K., Craigs, C., & House, A. (2012). Individual influences on lifestyle change to reduce vascular risk: A qualitative literature review. *British Journal of General Practitioners*, 62, e403–e410. doi:10.3399/bjgp12X649089
- Quant review of CVD risk communication - Sheridan, S., & Crespo, E. (2008). Does the routine use of global coronary heart disease risk scores translate into clinical benefits or harms? A systematic review of the literature. *BMC Health Services Research*, 8, 60. doi:10.1186/1472-6963-8-60
- Review of best practice risk communication for decision aids - Trevena, L. J., Zikmund-Fisher, B. J., Edwards, A., Gaissmaier, W., Galesic, M., Han, P. K. J., . . . Woloshin, S. (2013). Presenting quantitative information about decision outcomes: A risk communication primer for patient decision aid developers. *BMC*

	Medical Informatics and Decision Making, 13, S7. doi:10.1186/1472-6947-13-S2-S7 METHODS Is your specific patient population/context different from previous research, or does your interview schedule include a new angle? Did you intend to focus on people already taking medication, or did you try to sample more patients not on medication? What about experience of CVD events (heart attack/stroke)? Why did you develop a new decision aid instead of using one of the many existing CVD decision aids? Did you base your example on an existing decision aid? Analysis is very clearly explained RESULTS Don't cite/compare to other research in the results section – save for discussion Good use of quotes to illustrate subthemes Can you name your main themes more clearly and include this in your abstract, to more clearly outline the main findings? This may help with the discussion as well. Figures are helpful DISCUSSION What is new about your findings in comparison to qualitative research on CVD risk since 2008? Need to relate figure 2 and 3 to other literature in discussion more explicitly, see list above Clear explanation of limitations
--	---

VERSION 1 – AUTHOR RESPONSE

Reviewer 1.

In the title, the reason why the authors used the expression: People like you? is not evident. In addition, some information about the type of population included in the study should be appropriate (hypertensive patients).	We would prefer to keep the title if possible, which is explained within the abstract. This refers to the common phrase “out of 100 people like you” which is commonly used in decision aids. The key finding of our study is that participants did not widely identify the this population, and hence that the phrase may be counterproductive. We have added quotation marks around the phrase to improve clarity.
---	---

	We have added that the study considered people with hypertension in an edited title. We ask if the editors might reconsider their request around the title with this explanation.
Is the research question or study objective clearly defined? :No The authors describe what seem(s) to be the objective(s) of the study in the last part of the introduction through three groups of questions (first, page 4, lines 27-31; second, page 4, lines 34-39; third, page 4, lines 39-44), which are connected using logical links. Therefore, the reader may not directly apprehend which are the primary or secondary objectives of the study.	The aim of our study (to understand the perspectives of people with hypertension on future CVD risk) is described clearly, and in multiple places (including the title). The lines noted by the reviewer (the ‘three groups of questions’) expand upon the aim, and give (non-exhaustive) examples of the type of information which might be relevant from a qualitative study on this topic. We do not think it makes sense to define ‘primary’ and ‘secondary’ objectives. Given the design of the study (a qualitative interview, using a modified-grounded theory approach), it is perfectly appropriate to set an overall question, and for the precise answers to be defined by the data collected, rather than narrowly pre-specified.
Is the abstract accurate, balanced and complete?: No In the results section, the themes identified in the analysis of the data are not indicated, as reported in Methods/Analysis (page 6, lines 21-57) and the results are presented as a narrative, without a clear framework as commonly performed in qualitative studies.	We have improved the clarity of the reporting, by making the headers in the Results section match the key findings. We have additionally moved the Figure of the ‘lay model’ to the top of the Results, which provides a clearer framework for the thematic analysis below.
Are the methods described sufficiently to allow the study to be repeated?: No 1-The semi-structured interview schedule piloted and refined by the researchers is not provided (though indicated its existence in the online Appendix).	We apologise for this omission, we overlooked this document on the original submission. We have supplied the interview schedule with this resubmission.
2- Some criteria of COREQ guidelines for reporting qualitative studies are lacking such as: previous knowledge between the interviewer and the participant, duration of the interviews, exposure of the transcript to the interviewer for comments or corrections and features of candidates who refused to participate in the study.	We have added some details on these points as requested: “The recruitment sites were chosen to ensure that the interviewer and participants would have no prior knowledge of one another.” “ We did not seek further feedback from participants on the transcripts, other than seeking clarification of any unclear points during the course of the interview.” “To meet the requirements of the ethics committee, initial contact had to be from the

	patient's own doctor, and we do not have access to data on the characteristics of those who declined to participate/did not respond at this stage."
Are research ethics (e.g. participant consent, ethics approval) addressed appropriately?: No The approval by ethic committees is described (page 5, lines 14-19) but whether / how the participant provided informed consent is not specified.	We have added the following description: "Participants were invited to participate voluntarily, provided with study information to read, and asked to sign a written consent form if they agreed to participate."
Do the results address the research question or objective? No The research question/s was/were ill-defined and therefore, it is hard to say whether the results address them properly. In particular, the authors indicate that "the transcripts of the interviews were analysed using a form of grounded theory where analytical categories are developed inductively from the data rather than defined beforehand". Codes and themes extracted from the transcripts should answer predetermined research questions related to the main topic of the study which is "cardiovascular risk". Instead, the thematic analysis encompasses some aspects not related to the "patient perspectives on cardiovascular risk", such as, for example, causation of hypertension.	We use a grounded theory-based approach for analysing the data, which is commonplace for this type of study. This method does not recommend coding with a narrowly pre-specified question in mind. The approach we used for coding (creating new codes inductively from the data) is conventional for this type of research (the name 'grounded theory' refers to the analysis being 'grounded' in the data). As an example of why this is important, in a previous study we found that patient understanding of hypertension causation was an important factor in whether they decided to continue with medical treatment. If we were to have limited coding to a narrow question (in our previous study 'Why do patients often not take medication?') we would have missed this insight. We do agree that the section on hypertension causation is less relevant to the focus of our current analysis, and have amended it to focus on the consequences rather than the causation of hypertension.
Are they presented clearly? No 1- In page 7, lines 44-51, the authors describe the level of literacy of participants about causes of hypertension and not their perception of the risk that overt hypertension entails. This information might have been interesting if the impact of literacy on the perception of risk had been evaluated.	We agree that this original section was less relevant to the overall thematic synthesis. We have changed the emphasis of this section to perspectives on the consequences of hypertension (rather than the causation), and replaced the quotation with a more relevant example. We have additionally added the explanation that the same factors which were seen as causing hypertension, were seen as causative of complications of hypertension.
2- Some of the meanings derived from the transcripts do not seem to reflect adequately the participants' words. For example, in page 9, lines 29-37, the authors' interpretation is that "a key benefit of maintaining good health was that (they) could avoid thinking about future risk", whereas the actual expression of the participant	We thank the reviewer for raising these examples. For the first example (page 9 lines 29-37), we inadvertently transposed two adjacent quotes. We have replaced the quote illustrating the first concept. The reviewer is correct that the original

suggests that he/she has other medical problems that make futile for him/her the risk of complications of hypertension. Another examples are found in page 10, lines 19-21 and in lines 37-39 which actually do actually not depict reluctance to consider future risk. Indeed, the first participant, right or wrong, considers that he/she has a low chance of stroke (therefore, he already considered that risk) while the second one is expressing at least two potential consequences of hypertension (heart damage and stroke). Conversely, reluctance to risk estimates is described out-of-place in the paragraph of “Response to example risk charts” (page 11, lines 7-11).	quote illustrates CVD having a lower priority than other chronic health problems, and we have amended the description to reflect this. For the second example (page 10, lines 19-21), we describe that this quote illustrates that participant did not “need to contemplate [risk] further” (emphasis added), which we believe is a reasonable interpretation. The reviewer is correct that the title of the theme “reluctance to consider future risk” does not fit this well, and we have changed this to “Attitudes towards future risk of illness”, which covers people who not only avoiding contemplation of risk, but also those who feel no need to contemplate risk future. We have moved the quote in lines 37–39 up to the theme “Causes and consequences of hypertension” Regarding the last example (page 11, lines 7–11), we differentiate between perspectives on risk (i.e. how participants perceive their future risk of CVD?) versus perspectives on risk communications (i.e. participants responses to discussions around the probabilities as presented in an example decision aid). We have changed the section headers to make this clear.
3- It is appropriate to transcribe verbatim the participants’ words. However, it may be somewhat forced to obtain meaning units from uncomprehensible sentences such as, for example, that in page 13, lines 21-23but it kind of goes in phases (?)... In other words, the authors would select for the analysis only phrases that convey a well identifiable meaning.	We believe that this sentence makes sense. The participant is describing that his alcohol consumption “goes in phases” — i.e. is usually low, but occasionally high.
Are the discussion and conclusions justified by the results: No In conclusions, the idea proposed by the authors to engage patients in discussions about risk of using tools that incorporate risk factors for complications of hypertension which are regarded important to patients may be interesting but not real. In fact, available risk algorithms do not include some features, hardly measurable, that patients may consider relevant, such as family history, exposure to stress, dietary habits, etc. In addition, though It is not obvious how could an estimate of risk could be	We believe the reviewer is making the point that creating a CVD risk model which takes into account dietary habits and stress would be challenging, since these variables are difficult to collect. We agree with this point, and the Dundee Heart Rank system we describe uses conventional biomedical risk factors. We have added this text to clarify: “One disadvantage is that this approach would still ultimately rely on conventional risk models

objectively and scientifically provided by comparing a subject's risk with that of his peers or family members, which should be itself estimated somehow.	(which arguably still would not include the variables participants hold to be most important). Nonetheless, such a presentation could overcome some of the comprehension issues identified in this study, and is worthy of further investigation.” The Dundee Rank compares an individuals risk against an estimate of the general population of the same age group. The reviewer is correct that this is estimated (via the same underlying model, based on individual risk factors measured in a population survey). This is described in the manuscript: “The queue position was determined by reference to the risk factors of 10,359 men and women aged 40–59 in the Scottish Heart Health Study.”
Is the supplementary reporting complete (e.g. trial registration; funding details; CONSORT, STROBE or PRISMA checklist)? No The authors did not mention the utilization of any checklist for reporting qualitative research such as COREQ (COnsolidated criteria for REporting Qualitative research). In addition, as indicated previously, the interview scheduled was not provided.	We did not use a checklist in the original submission, but have agreed to submit the SRQR checklist as requested by the journal with this resubmission.
Is the standard of written English acceptable for publication? No There are many phrases throughout the text without an acceptable written English. Some of them are actually unintelligible. Therefore, we suggest the authors to subject the manuscript to a mother tongue scientific writer check before re-submission to any journal.	We have fixed a couple of typos, including one identified by the editor. We do not agree that parts of the article are unintelligible. However, if there are any specific examples where our writing could be improved we would be happy to fix them.

Reviewer: 2

Reviewer Name: Yeunjung Kim

Institution and Country: Yale University School of Medicine. CT, USA.

Competing Interests: none declared

This manuscript focuses on the much-needed topic of decision aids and understanding of personal CV risk and future risk. There were many aspects of the study which were enlightening and fortifying known data.	We thank the peer reviewer for their comments.
I do appreciate the qualitative aspect of the study and I believe authors have intimately detailed the discussions between the interviewer and interviewees. They are using a verified and stringent method for building a narrative of the	We are grateful for the reviewers comments about our description of the interviews, and methods used. We agree our original submssion did not

qualitative data. The OSOP method is described in general but it is unclear exactly how data saturation was met. This process could be a bit more transparent because the point of data saturation can definitely impact the outcome of a study this size.	adequately describe how data saturation was decided upon. We have added the following text to our methods section to address this: “Interview transcripts, notes from ongoing interviews, and the emerging analysis were reviewed regularly at meetings of the investigators (IM, CW, and CMcK) where the decision about data saturation was made, defined as when no new themes appeared to be generated from interviews.[15]”
In summary, at this time there is a lack of integration of qualitative findings to provide a rich narrative mostly due to the structure of the presentation. I think that the manuscript could be better organized to present the rich details of the interviews. There are few errors in the manuscript/abstract which can easily be fixed.	We agree that our original submission would have benefited from a better structure. To address this problem, we have made the following changes:  1. We have moved up the ‘lay model’ figure to the start of the Results section. 2. We have rewritten the introductory text to the Results section to emphasise the main themes. 3. We have made headers in the Results text which match the main themes identified. We believe this change in the presentation better integrates the findings.
- Patient characteristics: How about including comorbidities which may be associated with poor outcomes from hypertension (such as diabetes or coronary artery disease)?	As stated in the methods we excluded participants with pre-existing cardiovascular disease, since we wanted to understand the perspectives of people who had not personally experienced complications. We have clarified in the text that we also excluded people with diabetes and secondary hypertension.
- Table 1 could be more detailed (i.e. add education level and/or socioeconomic status). Table 1 could show the proportion/mean of each patient characteristics (i.e. mean age, race, socioeconomic status, education level, etc.)	We present summary data of participants age, ethnicity, and occupations in the ‘Characteristics of participants’ (including proportions). We have considered including more detailed information about individuals in Table 1, but examining the data we believe there is a risk that participants could be individually identified. We have therefore presented more limited data in the Table, and richer summary data in the text.
- Rather than listing simple ideas, if ideas can be grouped into certain overarching themes or concepts, this would be much more effective. (This can be applied throughout the results section.) For example, the “lay model” of	We are grateful for this comment and idea, and have followed the reviewers suggestion. We have additionally edited the Figure and headers in the Results section, so that they are presented as key themes.

cardiovascular risk could be presented initially and supported in the results section with the participant quotes.	
- Please integrate and present overarching findings. For example, there was one patient who felt reluctant to consider future risk because of “a happy life.” Rather than just having a quote saying that, by providing a more complete detail about the patient’s personal background, the readers may understand why he would say that (aside from being of older age). I think it would make the overall paper stronger. This is a general to the paper as a whole.	We thank the reviewer for their helpful suggestion. We have edited the presentation of a number of quotes throughout to give a better idea of the context, and the participants behind them. For the example given, we have provided some more description of the participant, and that the “happy life” comment was made by a man in his eighties, who was very physically active and in good health.
- Regarding the cause of hypertension. Unfortunately, much of the qualitative responses appear to be short answers rather than detailed narratives. At least, this is how the authors have represented the results. In this case, it may be more valuable to quantify the responses. I wished for more details of the respondents for which contextualization was done. For example, in attributing hypertension to new environment and stress, were there similarities of the participants aside from being born outside of the UK? Were the participants diagnosed with hypertension in the UK or prior to coming to the UK? Is there any association between how patients answered this question and how they applied the future CVD risk from hypertension?	The quotes presented are inevitably brief due to the presentation in a journal article, but as above, we have made changes to the presentation to give the narrative better flow. We have provided more context about the individuals who made the quotes around their perceived health, age, and background where it helps understand their interview. The principal themes did not obviously differ by age, country of origin, or ethnicity, and we have added some description of this to the Discussion section.
The vague descriptors (such as more than half, several, most, large portion, around half, many) were distracting especially when it was done in succession (i.e. in the paragraph regarding stress as a contributor to hypertension: First sentence says more than half of the participants. The next sentence says for some of these participants. The next sentence starts for several participants.)	We avoid providing exact numbers of respondents for a particular theme, as described in Ziebland et al. The main reason is that this design of qualitative study aims to describe a wide range of perspectives (the sampling is purposive rather than statistical). The existence of a theme in a certain proportion of the participants does not indicate the frequency in the general population, and themes which occur even in a single participant can be important. We have edited some of these descriptors to make them more consistent and easier to read. Ziebland, Sue, and Ann McPherson. 2006. “Making Sense of Qualitative Data Analysis: An Introduction with Illustrations from DIPEX (personal Experiences of Health and Illness).” Medical Education 40 (5): 405–14.
- Regarding the reluctance to consider future	We agree with this comment. As per our

risk. This was especially an important finding. As we know, the ability to perceive risk may be important in modifying behavior. Please define how “many participants expressed desire to avoid contemplating future risk.” Who were these people versus participants who were more engaged in considering future risk? Here, you may say that the avoidance was due to 3 themes: 1) inability to see the urgency of the risk, 2) purposeful neglect by focusing on the positive, and 3) belief that complications are unavoidable. This may be a stylistic difference in approach but helps us, the reader, to categorize the responses early on. So I believe having the key components of the “lay model” described earlier and supported would be stronger format than using it as a basic summary figure in the end.	response above, we have moved the “lay model” figure to the start of the Results section. We have additionally added new text early in the results section summarises the key themes. This text makes clearer some of the key findings: the avoidance of risk, and the explanations behind the avoidance.
- Regarding response to example risk charts. The authors can be more direct when describing more quantitative aspects of the study. For example, it would be more clear if authors say this # of participants found risk charts helpful and not just say around half. And who are these people and how are they similar or different to those who disagreed?	As described above, we have provided more detailed descriptions of the respondents throughout the Results section to provide better context. We have avoided providing exact numbers for the reasons given above. We have the additional issue in this example that we did not ask a simple yes/no question around whether the chart was useful, but rather interpreted patient narratives. There is (necessarily) some subjectivity in these judgments, and therefore ‘around half’ is a more accurate representation.
Discussion and conclusion. - I thought the discussion was very interesting and sets up value in this study. I do believe that patients are actually correct when they find the denominator irrelevant to them. Although using probabilities come naturally to those in academics or in medicine, it makes sense how it may not be intuitive for some. Especially, this may actually impact how patients will perceive his or her risk. Using familiar examples (i.e. friends, family, sibling) makes it relevant to the individual. The inclusion of the ranking model, therefore, is appropriate in the conclusion.	We are grateful for the reviewer’s comment.

Reviewer: 3

Reviewer Name: Carissa Bonner

Institution and Country: University of Sydney, Australia

Competing Interests: None declared

Thank you for the opportunity to review this qualitative paper on CVD risk perceptions amongst hypertensive patients. The paper is generally well written (but with some minor grammatical errors throughout – please read carefully before next submission), describes the analysis methods very clearly, and provides some interesting findings and conclusions. However the paper needs to include much more explanation of the existing qualitative literature on this topic since the cited 2008 review, and how this study addresses a gap or adds to the literature. Is your specific patient population/context different, or does your interview schedule include a new angle? How do your findings relate to other qual studies and reviews on CVD risk perception, CVD decision aids, and recommended risk communication formats for decision aids? Specific suggestions/comments are below.	We are grateful for the reviewer’s comments. We have substantially expanded our discussion of qualitative studies done after the 2008 review in both the Background and Discussion section, and how our study fits in. Although there is a degree of overlap between our study and other qualitative studies on CVD risk, the key difference is our focus on CVD risk in the sense of likelihood (rather than risk factors or studies of understanding of disease causation). We have fixed a small number of typos in the manuscript.
STRENGTHS AND LIMITATIONS BOX Re: “Although many qualitative studies have examined patient perspectives on hypertension, this is one of few studies to examine perspectives on future cardiovascular risk” - There is a lot of literature on future CVD risk perceptions not cited so I don’t think this statement really reflects a key strength of the study – many studies looking at this issue include a more comprehensive range of risk factors for CVD (e.g. high cholesterol and/or lifestyle risk factors as well as hypertension) so may have been missed because of that – but they are still relevant.	We agree with the reviewer. Our original submission neglected key papers which examined perspectives on CVD risk. We have reworded the first point in the box to better describe how this study fits in: • We conducted a qualitative study to investigate how people with hypertension understand their future risk of developing cardiovascular disease. We have additionally added substantially to the Background and Discussion sections around other relevant studies.
INTRODUCTION How is hypertension defined in this context (thresholds vary by country/guidelines), and why have you chosen to focus on this single risk factor for CVD and not a more comprehensive range of risk factors for which blood pressure medication may be prescribed?	We wanted to understand the experiences of people with hypertension, since they would have been diagnosed with a risk factor (which is typically asymptomatic). The reviewer is correct that we could equally have chosen a variety of other related conditions for which blood pressure medication is prescribed.

There is a lot of qualitative research on the topic of future CVD risk perception since the 2008 review cited, which should be discussed, not just the research cited in South Wales. I suggest looking at the reviews below and many papers in the journal Patient Education and Counseling for a large literature on this topic. Some examples you might look into include:

EXAMPLES OF OTHER QUALITATIVE STUDIES ON FUTURE CVD RISK PERCEPTION

- Qual study in Netherlands – Damman, O. C., & Timmermans, D. R. M. (2012). Educating health consumers about cardio-metabolic health risk: What can we learn from lay mental models of risk? Patient Education and Counseling, 89,300–308.
- Qual study in UK - L. Polak, J. Green, Using quantitative risk information in decisions about statins: a qualitative study in a community setting, Brit. J. Gen. Pract. 65 (2015) e264–269, doi:<http://dx.doi.org/10.3399/bjgp15X684433>.
- Qual study in Australia – Bonner C, McKinn S, Lau A, Jansen J, Doust J, Trevena L, McCaffery K (2018). Heuristics and biases in cardiovascular disease prevention: How can we improve communication about risk, benefits and harms? Patient Education and Counseling, 101(5), 843–853. doi:
<https://emea01.safelinks.protection.outlook.com/?url=https%3A%2F%2Fdoi.org%2F10.1016%2Fj.pec.2017.12.003&PubMed&data=01%7C01%7Ciain.marshall%40kcl.ac.uk%7Ce5d5eee1c80b4bf89ae108d5d76da5f9%7C8370cf1416f34c16b83c724071654356%7C0&sdata=MZ5NIFS5VKqIVM%2BX%2BX24G%2FE3E2%2BFinjHG1oxk7tuGJQ%3D&reserved=0>

RELEVANT REVIEWS ON CVD RISK PERCEPTION (INCLUDING QUALITATIVE STUDIES)

- Review of CVD risk communication formats including some qual studies - T. Waldron, S. van der Weijden, J. Ludt, G. Gallacher, What are effective strategies to communicate cardiovascular risk information to patients? A systematic review, Patient Educ. Couns. 82 (2011) 169–181, doi:<http://dx.doi.org/10.1016/j.pec.2010.04.014>.
- Qual review on reducing CVD risk -

We agree with the reviewer. The Background section in the original submission focused on the research in South Wales, but neglected other relevant qualitative studies. We have made substantial edits to discuss other relevant research.

We are most grateful to the reviewer for taking the time to compile these references, and in particularly the relevant qualitative studies.

We have added to our Background and Discussion sections to incorporate discussion of all of the qualitative studies. We have additionally added references to the reviews by Waldron and Trevena to support our discussion section.

Murray, J., Honey, S., Hill, K., Craigs, C., & House, A. (2012). Individual influences on lifestyle change to reduce vascular risk: A qualitative literature review. British Journal of General Practitioners, 62, e403–e410. doi:10.3399/bjgp12X649089  • Quant review of CVD risk communication - Sheridan, S., & Crespo, E. (2008). Does the routine use of global coronary heart disease risk scores translate into clinical benefits or harms? A systematic review of the literature. BMC Health Services Research, 8, 60. doi:10.1186/1472-6963-8-60 • Review of best practice risk communication for decision aids - Trevena, L. J., Zikmund-Fisher, B. J., Edwards, A., Gaissmaier, W., Galesic, M., Han, P. K. J., . . . Woloshin, S. (2013). Presenting quantitative information about decision outcomes: A risk communication primer for patient decision aid developers. BMC Medical Informatics and Decision Making, 13, S7. doi:10.1186/1472-6947-13-S2-S7 	
METHODS Is your specific patient population/context different from previous research, or does your interview schedule include a new angle?	Our study has a degree of overlap with previous studies, but with a difference in focus. The primary aim of our study was to examine perspectives on perspectives on CVD risk (i.e. the probability, or uncertainty) rather than on risk factors (i.e. explanation of causation). The studies by Davison et al. and Demmer et al. primarily focused on explanations of disease causation, and the understanding and priority given to risk factors. The study by Demmer et al. does discuss that participants often saw risk as dichotomous, and we have added discussion of this interesting similarity with our own study to the Discussion. There is also some overlap in that understanding of CVD causation is important factor which influences perspectives on risk (and we also cover this topic in our interviews and analysis). However, the difference in focus is reflected in the much larger space given to discussions of risk (in the sense of likelihood) in the interview, analysis, and final manuscript compared with other related studies. We have added a footnote to explain this distinction, and also expanded our discussion of how our study fits with other related studies in the Background and Discussion sections.

Did you intend to focus on people already taking medication, or did you try to sample more patients not on medication? What about experience of CVD events (heart attack/stroke)?	We did not include whether people took medication or not in our sampling. All but one participant was prescribed medication, but the group were highly heterogeneous in their medication taking, ranging from those who took their medication by the clock to those who rarely used it. We have added some more detail to this description in the manuscript. We excluded people who had previous experience of CVD events personally, since we expected that these people would have very different ideas of CVD risk. We did report this in the original manuscript but have clarified the wording.
Why did you develop a new decision aid instead of using one of the many existing CVD decision aids? Did you base your example on an existing decision aid?	We were primarily concerned with the concept of risk in general, rather than a response to a particular decision aid. The decision aid was used primarily as a prompt to wider discussion rather than for detailed scrutiny. We therefore wanted to avoid any irrelevant distracting information (for purposes of our study), such as information on treatment options. This contrasts with the study by Bonner et al. who did a 'talk-aloud' study considering how patients responded to specific tools. However, most CVD decision aids strongly resemble one-another, and contain the same core components as our example (some text with risk communicated natural frequencies with a common denominator, plus an icon array). Our example followed the guidelines from IPDAS for communicating quantitative information. We have added explanatory text about how the decision aid was used in the interviews.
Analysis is very clearly explained	We thank the reviewer for their comments.
RESULTS Don't cite/compare to other research in the results section – save for discussion	Thank you, we have removed the reference to previous research.
Good use of quotes to illustrate subthemes	
Can you name your main themes more clearly and include this in your abstract, to more clearly outline the main findings? This may help with the discussion as well.	We agree, and have made substantial edits to the results presentation. We have moved the figure to the start of the Results section, together with new text summarizing the key themes from the analysis. We have additionally changed the headers in the

	Results section to match the main themes.
Figures are helpful	
DISCUSSION What is new about your findings in comparison to qualitative research on CVD risk since 2008?	We have substantially revised the Discussion section to incorporate more recent qualitative research on risk, and how our study fits in.
Need to relate figure 2 and 3 to other literature in discussion more explicitly, see list above	As above we have expanded our discussion of relevant literature in both the Background and Discussion sections. We have included descriptions of relevant qualitative studies to the 'lay model' Figure, and additionally provided discussion around the literature on the use of ranking systems for communicating risk.
Clear explanation of limitations